# Antileukemic Cell Proliferation of Active Compounds from Kaffir Lime (*Citrus hystrix*) Leaves

**DOI:** 10.3390/molecules25061300

**Published:** 2020-03-12

**Authors:** Songyot Anuchapreeda, Fah Chueahongthong, Natsima Viriyaadhammaa, Pawaret Panyajai, Riki Anzawa, Singkome Tima, Chadarat Ampasavate, Aroonchai Saiai, Methee Rungrojsakul, Toyonobu Usuki, Siriporn Okonogi

**Affiliations:** 1Department of Medical Technology, Faculty of Associated Medical Sciences, Chiang Mai University, Chiang Mai 50200, Thailand; fahmyfah@hotmail.com (F.C.); fai.natsima@gmail.com (N.V.); panyajaip@gmail.com (P.P.); singkome@gmail.com (S.T.); 2Department of Materials and Life Sciences, Faculty of Science and Technology, Sophia University, Tokyo 102-8554, Japan; r-anzawa-4g3@eagle.sophia.ac.jp; 3Cancer Research Unit of Associated Medical Sciences (AMS CRU), Faculty of Associated Medical Sciences, Chiang Mai University, Chiang Mai 50200, Thailand; 4Research Center of Pharmaceutical Nanotechnology, Chiang Mai University, Chiang Mai 50200, Thailand; 5Department of Pharmaceutical Sciences, Faculty of Pharmacy, Chiang Mai University, Chiang Mai 50200, Thailand; aimchadarat@windowslive.com; 6Department of Chemistry, Faculty of Science, Chiang Mai University, Chiang Mai 50200, Thailand; saiai_aroonchai@hotmail.com; 7College of Alternative Medicine, Chandrakasem Rajabhat University, Bangkok 10900, Thailand; mathewhor@hotmail.com

**Keywords:** kaffir lime, phytol, lupeol, Wilms’ tumor 1, leukemia, antiproliferation

## Abstract

Kaffir lime (*Citrus hystrix*) is a plant member of family Rutaceae, and its leaves are commonly used in folk medicine. The present study explores antileukemic effects of the extracts and purified active compounds from the leaves. The antileukemic activity was investigated via inhibition of Wilms’ tumor 1 (WT1), which is a protein that involves in leukemic cell proliferation. In addition, the compounds were investigated for their effects on *WT1* gene expression using real time RT-PCR and Western blotting. Cell cycle arrest and total cell number were investigated using flow cytometry and trypan blue exclusion method, respectively. The results demonstrated that the hexane fractionated extract had the greatest inhibitory effect on *WT1* gene expression of many leukemic cell lines and significantly decreased WT1 protein levels of K562 cells (representative of the leukemic cells), in a dose- and time-dependent manner. Subfraction No. 9 (F9) after partial purification of hexane fractioned extract showed the highest suppression on WT1 protein and suppressed cell cycle at G2/M. The organic compounds were isolated from F9 and identified as phytol and lupeol. The bioassays confirmed antiproliferative activities of natural products phytol and lupeol. The results demonstrated anticancer activity of the isolated phytol and lupeol to decrease leukemic cell proliferation.

## 1. Introduction

Kaffir lime, a common name of *Citrus hystrix*, is a plant belongs to the Rutaceae family and widely grown in Thailand and known as Makrut. It is also widely distributed and cultivated in many countries including Southeast Asia. The leaves of the kaffir lime are dark green color with a glossy sheen. Kaffir lime leaves and peels are aromatic and used as spices for various flavoring purposes such as seasoning or preparing savory curry pastes (red and green curry recipes). As a traditional medicine in Thailand, the leaves are used to maintain healthy teeth and gums, and as a remedy for scurvy. The leave extracts were reported to have antioxidant, anti-cancer, and anti-inflammatory activities [1]. Meanwhile, the juice is used to clean the blood, dispel gas, increase appetite, and to keep the hair and scalp in good health. Lawrence et al. reported that the main chemical constituents in kaffir lime peel were β-pinene (30.6%), limonene (29.2%), and sabinene (22.6%), and the main compound in leaves was citronellal (65.4%) [2]. However, reports concerning the amount of each chemical content were slightly different. For example, Chantaphon et al. reported that β-pinene (30.48%), sabinene (22.75%), and citronellal (15.66%) as the major components were extracted from the hydro-distilled essential oil of kaffir lime peel [3]. Kasuan et al. also showed that the steam-distilled essential oil of kaffir lime included limonene (27.97%), citronellal (15.31%), and α/β-pinene (9.82%) in the case of the peel and citronellal (71.55%) in the case of the leaf [4]. Waikedre et al. found that the main constituents of the essential oil of kaffir lime leaf grown in New Caledonia were also monoterpenes with terpinen-4-ol (13.0%) and β-pinene (10.9%) as the principal compounds [5]. In addition to the hydro-distillation technique, essential oil from the peels and leaves of kaffir lime may be extracted using solvent extraction. The major constituents of the ethyl acetate extracts from the kaffir lime peel were found to be limonene (31.64%), citronellal (25.99%), and β-pinene (6.83%) [2].

Many studies have demonstrated various biological activities of kaffir lime. The peel and leaf are sources of phenolic compounds [6] and antioxidative substances [6,7]. Furthermore, the extracts from the leaf and the peel possess the strongest effect as regards protection of deoxyribose from OH, suggesting the free-radical scavenging and anti-inflammatory activities of kaffir lime [8]. Most of the studies on kaffir lime bioactivities are associated to its antimicrobial effect. The essential oil and the extracts also exhibit antimicrobial activities [5,7]. Limonoids, a family of triterpenoids with putative anticancer properties in citrus fruits, exert a strong multifaceted lethal action against human neoblastoma and colon cancer cells [9]. The volatile oil of *Citrus aurantifolia* containing two major compounds, D-limonene and D-hydrocarvone, can inhibit the proliferation of colon cancer cells by apoptosis mediation [10]. Flavones, active compounds detected in *Citrus reticulata*, were found to have in vitro anti-tumor activity against human promyelocytic leukemic cells and murine myeloid leukemic cells [11]. The two glyceroglycolipids, 1,2-di-*O*-α-linolenoyl-3-*O*-β-galactopyranosyl-*sn*-glycerol and a mixture of two compounds, 1-*O*-α-linolenoyl-2-*O*-palmitoyl-3-*O*-β-galactopyranosyl-*sn*-glycerol and its counterpart, extracted from kaffir lime leaves, were found to be potent inhibitors of tumor promoter-induced Epstein-Barr virus (EBV) activation and 12-*O*-tetradecanoylphorbol 13-acetate (TPA), a skin carcinogen, activities in mice [12]. The essential oils of kaffir lime leaf and peel prepared by steam distillation was reported to have anti-proliferative activity on KB (cervical cancer) and P388 (mouse leukemia) cell lines [13]. The crude ethanol extracts of kaffir lime leaf and peel showed cytotoxic effects on K562, Molt4, U937, and HL60 cell lines, while the crude kaffir lime leaf extract demonstrated strong cytotoxic effects on human leukemic cell lines, and were non-toxic to normal peripheral blood mononuclear cells (PBMCs) [14]. Our previous study demonstrated that the fractionated crude extract of kaffir lime leaves obtained from hexane, ethanol, ethyl acetate, *n*-butanol, and methanol layer were investigated for their potential cytotoxic activity on K562, Molt4, U937, and HL60 cell lines, using the MTT assay. The ethyl acetate fractionated extract exhibited the highest cytotoxicity, with IC_50_ values of 35.3±1.4, 21.8±0.4, 19.8±1.0, and 19±0.6 µg/mL, respectively [15].

Cytotoxic effects of the ethanol extract and essential oil from kaffir lime leaf on leukemic cells have been reported in previous studies. In the present study, we investigated the most effective kaffir lime leaf fractionated extract and the elucidated active compounds on the human Wilms’ tumor (*WT1*) gene expression in leukemic cells. The human Wilms’ tumor (*WT1*), the gene encoding the zinc finger transcription factor, is important for cell survival, differentiation, and proliferation. In recent years, the study of WT1′s involvement in malignant cells has unexpectedly revealed a potential role for WT1 as an oncogene, especially in leukemia [16]. WT1 is highly expressed in the bone marrow or the peripheral blood of a variety of leukemia in comparison to normal bone marrow and normal progenitor cells [17,18]. The *WT1* gene is highly expressed in various types of leukemias including acute myeloblastic leukemia (AML), acute lymphoblastic leukemia (ALL), and chronic myelocytic leukemia (CML) and is important for leukemia treatments, progression, and prognosis [16,19]. WT1 signaling pathway in leukemic cells has been previously revealed to involve protein kinase Cα (PKCα) and c-Jun N-terminal kinase (JNK) proteins in K562 cells [20]. Furthermore, AP-1 has been reported to contribute to WT1 autoregulation of *WT1* gene expression in K562 cells [21]. Curcumin was firstly reported to inhibit WT1 protein expression by PKCα suppression and decrease leukemic cell proliferation [20]. The goal of the present study is to provide new basic knowledge on the active compounds in kaffir lime leaf extracts that have antileukemic activity.

## 2. Results and Discussion

### 2.1. Yield of Kaffir Lime Leaf Extracts

In the present study, two kilograms of kaffir lime leaves were extracted using five organic solvents, including ethanol, hexane, ethyl acetate, *n*-butanol, and methanol, having relative polarities of 0.654, 0.009, 0.228, 0.602, and 0.762, respectively, compared to water (1.000), The yields of the extracts were 10.36, 3.51, 1.12, 2.78, and 10.31%, respectively. The highest yield was found to be the ethanol extract. It was considered that ethanol can dissolve many compounds present in kaffir lime leaves.

### 2.2. Yield of Fractions and Pure Compounds After Purification Process by Column Chromatography

Thirteen fractions were collected by partial purification using column chromatography. Fraction 1 was the original hexane fractionated extract. The dry weight and % yield of fraction 2 to 13 are presented in Appendix A. Subfraction No. 7, 8, 9, 10, and 12 had high percentages of yields, with values of 12, 12, 9.8, 8.9, and 12%, respectively. The purified phytol and lupeol after column chromatography of F9 was 8.1 and 6.6%, respectively. To identify the compounds, spectroscopic analyses, including mass spectrometry and ^1^H and ^13^C nuclear magnetic resonance (NMR), were performed. The structures of the two compounds are shown in Figure 1A,B. Phytol was first discovered as the alcohol portion of chlorophyll [22]. Lupeol was first described in 1891 [23].

### 2.3. Effect of the Extracts on WT1 mRNA Levels in K562, Molt4, U937, and HL60 Cell Lines

The objective of this experiment was to determine the effect of crude kaffir lime leaf extracts on the levels of WT1 mRNA in various types of leukemic cell lines. The IC_20_ values of ethanol, hexane, ethyl acetate, *n*-butanol, and methanol extracts were reported to be non-cytotoxic in four leukemic cell lines, K562, Molt4, U937, and HL60 by MTT assay after incubated for 48 h [15]. The IC_20_ values have been shown in Table 1. Non-cytotoxic dose is always used to study the effects of drugs or compounds on gene and protein expressions [24,25]. DMSO treatment was used as a vehicle control. The results are shown in Figure 2A. It was found that all extracts decreased the WT1 mRNA levels in the K562 cells by 49, 60, 24, and 5%, when compared to the vehicle control. The WT1 mRNA levels were found to have decreased by 22, 51, and 26%, respectively, after the crude ethanol, hexane, and ethyl acetate extract treatments, whereas the *n*-butanol extract was observed to have no inhibitory effect on the WT1 mRNA expression in the Molt4 cells, compared to the vehicle control. In the case of U937 cells, all extracts decreased the WT1 mRNA levels by 5, 42, 30, and 9%, respectively, compared to the vehicle control. All extracts decreased the WT1 mRNA levels in the HL60 cells by 55, 56, 34, and 48%, respectively, compared to the vehicle control.

The study of the effects of kaffir lime leaf fractionated extracts on the *WT1* gene expression in K562, Molt4, U937, and HL60 cell lines using non-cytotoxic doses of crude extracts at IC_20_ values suggested that all the crude extracts could decrease the WT1 mRNA levels. Nevertheless, it was observed that only the hexane extract had strong inhibitory effect on the *WT1* gene expression, and that the concentrations of the hexane extract used in the four leukemic cell lines were lower than those for the other crude extracts used in this study. Referring to the study on the cytotoxicity of crude kaffir lime leaf fractionated extracts, the hexane extract showed significantly high cytotoxic effect on the four leukemic cell lines as well. Thus, the results from the two experiments demonstrated that the active compounds dissolved in hexane fraction may have the ability to destroy leukemic cells at high doses and to downregulate the WT1 mRNA level at non-cytotoxic doses.

### 2.4. Effect of Concentrations and Contact Time of the Extract on WT1 mRNA Levels in K562 Cell Line

Based on the WT1 mRNA levels after the treatments, it can be inferred that the crude kaffir lime leaf hexane extract possessed extremely strong inhibitory effect on the *WT1* gene expression in the K562, Molt4, U937, and HL60 leukemic cell lines. The reduction in the WT1 mRNA expression was associated with decreased cell proliferation in the leukemic cells and leukemic cell lines (K562 and HL60) [26], suggesting that WT1 plays a role in leukemogenesis. In addition, different concentrations of hexane extract were used to study the effect on the *WT1* gene expression and a dose-dependent manner on leukemic cell lines. The K562 cell line was chosen as a representative of leukemic cell lines. The leukemic cell line was treated with the extract at final concentrations of 5, 10, 15, and 20 µg/mL (non-cytotoxic doses), and 0.08% DMSO was used as the vehicle control. After 48 h of incubation, the treated cells were harvested and extracted for determining the mRNA levels by real-time RT-PCR. The percentages of the WT1 mRNA levels were found to be 74.7 ± 11.4, 64.3 ± 4.0, 57.7 ± 2.5, and 52 ± 4.4% in response to the treatment with concentrations of 5, 10, 15, and 20 µg/mL, respectively, and it was observed that the hexane extract could decrease the WT1 mRNA levels in a dose-dependent manner by 25, 36, 42, and 48%, respectively, as compared to the vehicle control (Figure 2B).

In order to study the effects of contact time of the extract, the K562 cells were treated with 13.6 µg/mL (IC_20_) of hexane extract for 24, 48, and 72 h, respectively. The vehicle control (0.05% DMSO) was treated for 72 h. After incubation, the treated cells were harvested and extracted for determination of mRNA levels. The WT1 mRNA levels were found to be 81.7 ± 11.9, 62 ± 4.4, and 57.3 ± 4.9% in response to 24, 48, and 72 h, respectively. It was concluded that the hexane extract could decrease the WT1 mRNA levels in a time-dependent manner by 17, 38, and 43%, respectively, as compared to the vehicle control (Figure 2C).

### 2.5. Effect of the Extracts on WT1 Protein Levels in K562 Cells

According to the study carried out by Anuchapreeda et al., the WT1 protein was detected in only the K562 cells, whereas, the expression of WT1 could not be detected in the U937 and HL60 cell lines [27]. However, glyceraldehyde 3-phosphate dehydrogenase (GAPDH) was used as the internal control. To determine the effect of crude kaffir lime leaf extracts on the levels of the WT1 protein expression in the K562 cells, the leukemic cell line was cultured in a medium containing 0.18% DMSO (vehicle control) and crude kaffir lime leaf extracts from ethanol, hexane, ethyl acetate, and *n*-butanol, at the IC_20_ concentrations of 45.4, 13.6, 11.9, and 40.9 µg/mL, respectively, for 2 days. The percentages of the WT1 protein levels were found to be 58.7 ± 6.2, 38.9 ± 10.4, 83.4 ± 8.6, and 99.2 ± 9.5% in response to the ethanol, hexane, ethyl acetate, and *n*-butanol extracts, respectively. From this experiment, it was concluded that the extracts could decrease the WT1 protein levels in the K562 cells by 41, 61, 17, and 1%, respectively, when compared to the vehicle control (Figure 3A).

### 2.6. Effect of Concentrations of Extracts on WT1 Protein Levels in K562 Cell Line

In order to study the effect of hexane extract dose, the K562 cells were treated with 5, 10, 15, and 20 µg/mL of crude hexane extract, and 0.08% DMSO was used as the vehicle control for 2 days. The percentages of the WT1 protein levels after normalization with GAPDH were found to be 93.3 ± 12.9, 79.1 ± 5.7, 61.4 ± 9.2, and 36.8 ± 9.7% in response to 5, 10, 15, and 20 µg/mL, respectively. It was concluded that the hexane extract could decrease the WT1 protein level in a dose-dependent manner by 7, 21, 39, and 63%, respectively, compared to the vehicle control (Figure 3B).

### 2.7. Effect of Contact Time of the Extract on WT1 Protein Levels in K562 Cell Line

In this experiment, we studied the effect of hexane extract treatment on K562 cells with different treatment durations, the cells were treated with 13.6 µg/mL of hexane fractional extract for 24, 48, and 72 h, respectively. The vehicle control (0.05% DMSO) was treated for 3 days. The percentages of the WT1 protein levels after normalization with GAPDH were observed to be 92.9 ± 8.0, 65.9 ± 15.8, and 34.1 ± 28.4% in response to 24, 48, and 72 h, respectively. Thus, it can be concluded that the crude hexane extract could decrease the WT1 protein level in a time-dependent manner by 7, 34, and 66%, respectively, compared to the vehicle control (Figure 3C).

### 2.8. Cytotoxicity of Subfractions of Hexane Extract on K562 Cells Using MTT Assay

The K562 cells were treated with fractions after partial purification by column chromatography at various concentrations for 48 h, and the cytotoxic effects were investigated using the MTT assay. The results showed that subfraction numbers 1, 2, 9, 10, 11, and 12 had cytotoxic effects on the K562 cells, whereas subfraction numbers 3, 4, 5, 6, 7, 8, and 13 demonstrated no cytotoxicity (IC_50_>100 µg/mL). The IC_50_ values of the active subfractions (1, 2, 9, 10, 11, and 12) were found to be 40.8 ± 5.7, 86.1 ± 5.8, 20.8 ± 2.1, 42.8 ± 4.8, 53.1 ± 3.9, and 57 ± 4.2 µg/mL, respectively. Thus, subfraction numbers 1, 9, and 10 showed good cytotoxicity (Figure 4A).

### 2.9. Effect of Subfraction No. 1, 9, and 10 on WT1 Protein Levels in K562 Cells

Based on the cytotoxic tests, the most effective subfractions, subfraction No. 1, 9, and 10, were chosen to confirm their activities on the WT1 protein expression. The concentrations used in this study were IC_20_ values (17.8, 10, and 16.6 µg/mL, respectively). Subfraction No. 9 (F9) showed the highest inhibition on WT1 protein expression in the K562 cells, with an inhibitory value of 84.9%, while subfraction No. 1 and 10 were found to have inhibitory values of 39.4 and 40.7%, respectively (Figure 4B).

### 2.10. Effect of F9 on Cell Cycle Distribution in K562 Cell Line by Flow Cytometry

This experiment was to determine the effect of contact time of F9 on cell cycle distribution in K562 cells. The cells were cultured with F9 at the concentrations of 10 µg/mL (IC_20_ value) and DMSO (vehicle control) for 12, 24, 36, and 48 h and assessed by flow cytometric analysis after DNA staining with PI. The flow cytometry data at 12, 24, 36, and 48 h are shown in Figure 5A. It was clearly observed that after the cells were treated with F9, the cells were significantly arrested at the G2/M phase with the increasing of percent cell population of 41.4% as compared to the vehicle control (16.8%) at 24 h (Figure 5A–C). Sub-G1 (peak of apoptotic cells) was observed after F9 treatment for 48 h. However, the percent of cell death was less than 20%, as determined by trypan blue exclusion method (Figure 5C).

### 2.11. Effect of F9 on Cyclin B and p53 Protein Expressions in K562 Cell Line

K562 cells were treated with 10 μg/mL (IC_20_ value) of F9 and 0.04% DMSO for 12, 24, 36, and 48 h. F9 significantly decreased the cyclin B protein levels in a time-dependent manner at 24, 36, and 48 h by 22, 27, and 48%, respectively, when compared to vehicle control (Figure 6A). F9 treatment also decreased cyclin B expressions with increasing dosage (Figure 6B). F9 significantly increased p53 protein levels at 24 h by 26% when compared to vehicle control (Figure 6C); this increase in p53 protein expressions was also dose-dependent manner (Figure 6D). Furthermore, a marked decrease in total number of cells was observed with increasing incubation time and dose of F9 (Figure 6E,F). These results may be explained through changes in cellular signaling; a kinase signaling array was used to examine the phosphorylation/activation state of various key kinases. In this experiment, K562 cells were treated with non-cytotoxic doses of F9 (10 μg/mL) for 48 h. As shown in Figure 7, F9 strongly increased poly (ADP-ribose) polymerase (PARP). It also increased caspase-3, JNK, and p53 which are related to cell apoptosis and cell cycle arrest. JNK has been reported to be involved in cell apoptosis [28,29]. 

### 2.12. Effect of Phytol and Lupeol on Cytotoxicity and WT1 Protein Expression in K562 Cells

This experiment was performed to confirm the activity of the isolated compounds in F9. The activities of the isolated phytol and lupeol were tested and compared to F9 on cell cytotoxicity and WT1 protein expression in K562 cells. In the past, phytol and lupeol were found in chloroform fractionated extract of kaffir lime leaves [30,31]. Phytol has been reported to have cytotoxic effects on solid tumor cell lines i.e., MCF-7, MDA-MB-231, HeLa, PC3, HT-29, A-549, Hs294T, and MRC-5 with IC_50_ values ranging from 8.79 to 124.84 µM [32]. Lupeol has been reported to have anti-inflammatory and anticancer activities and is consumed as a dietary supplement [33]. In the current study, phytol and lupeol showed cytotoxic effects to the tested leukemic cells with IC_50_ values of 45.80 ± 4.47 and 82.8 ± 4.6 µg/mL, respectively. However, the cytotoxicity of these compounds was less than F9 (20.08 ± 2.1 µg/mL), by 2-fold for phytol, and 4-fold for lupeol (Figure 8A). IC_50_ values of known chemotherapeutic drugs for leukemia including doxorubicin (0.8 ± 0.06 µg/mL) and idarubicin (0.41 ± 0.04 µg/mL) were used to compare cytotoxicity of phytol and lupeol in K562 cells. Phytol showed less cytotoxicities than doxorubicin and idarubucin by 56- and 110-fold, respectively while lupeol showed less cytotoxicities than both drugs by 103- and 202-fold, respectively (Table 2). Moreover, the cytotoxicity of phytol and lupeol were less than vincristine (0.008 µg/mL) in the previous report by 5635- and 10,350-fold, respectively [14]. Furthermore, phytol (IC_20_ = 21 µg/mL) and lupeol (IC_20_ = 40 µg/mL) showed a significant decrease on WT1 protein expressions by 33.98 and 24.81% (*p* < 0.05) as shown in Figure 8B. F9 was more effective than phytol and lupeol with suppression value of 74.74%. Total cell numbers decreased by 24.18 and 29.41%, respectively, when compared to vehicle control while F9 treatment gave 52.29% decrease (Figure 8C). Phytol and lupeol are suggested to suppress *WT1* gene expression via WT1 signaling pathway that involved PKCα and JNK proteins in K562 cells [20,21]. 

## 3. Materials and Methods 

### 3.1. Chemicals and Reagents

RPMI-1640, penicillin/streptomycin, l-glutamine, secondary antibody conjugated with horseradish peroxidase (HRP), RNaseOUT™ Recombinant Ribonuclease Inhibitor, and trypan blue were purchased from Invitrogen™ (Grand Island, NY, USA). Fetal bovine serum (FBS) and penicillin/streptomycin were purchased from GIBCO-BRL (Grand Island, NY, USA). Protease inhibitors were purchased from Amresco^®^, Solon, OH, USA. SuperSignal^®^ West Pico Chemiluminescent was purchased from Pierce, Rockford, IL, USA. MTT (3-(4,5-dimethylthiazol-2-yl)-2,5-diphenyltetrazolium bromide), propidium iodide (PI), and standard phytol were purchased from Sigma-Aldrich (St. Louis, MO, USA). Dimethyl sulfoxide (DMSO), ethanol, ethyl acetate, hexane, *n*-butanol, and methanol were purchased from Labscan (Dublin, Ireland). Analytical thin layer chromatography (TLC) was performed on Silica gel 60 F254 plates produced by Merck (Darmstadt, Germany). Column chromatography was performed with neutral Silica gel 60N (spherical, 40-50 µm) produced by Kanto Chemical, Co. Ltd. (Tokyo, Japan). High Pure RNA Isolation Kit and Transcriptor High Fidelity cDNA Synthesis Kit were purchased from Roche Applied Science (Mannheim, Germany). DyNAmo™ qPCR Kit was purchased from FINNZYMES (Espoo, Finland). Tagman^®^ hydrolysis probe was purchased from Operon Biotechnology, (Huntsville, AL, USA). Rabbit polyclonal anti-WT1 antibody (C-19) and rabbit polyclonal anti-GAPDH antibody (FL-335) were purchased from Santa Cruz Biotechnology (CA, USA). Anti-mouse cyclin B IgG and anti-mouse p53 IgG were purchased from BD Biosciences™ (San Jose, CA, USA). RNase A was purchased from Invitrogen™ (Carlsbad, CA, USA). PathScan^®^ Intracellular Signaling Array Kit (Chemiluminescent Readout) was purchased from Cell Signaling Technology^®^ (Danvers, MA, USA).

### 3.2. Plant Material and Extract Preparation

Kaffir lime leaves were collected from Amphoe Muang, Chiang Mai Province, Thailand in June 2017. The leaves were extracted according to the previously described method with some modification [15]. Briefly, a half portion of dried kaffir lime leaf powder (476 g) was macerated in 95% ethanol for 1 day, and the liquid portion was collected. The filtrate was collected and then evaporated using a rotary evaporator (EYELA N-1000, Bohemia, NY, USA) and subsequently freeze-dried to obtain a crude ethanol extract. Another half portion was subjected to sequential maceration. First, it was macerated in hexane for 1 day and the liquid portion was collected. After that, the whole liquid portion was filtered. The filtrate was collected and then evaporated using a rotary evaporator to obtain a hexane extract. The residue marc after hexane extraction was placed under the hood to remove the hexane. The dried residue was further extracted with ethyl acetate, using the same procedure, to obtain ethyl acetate fractionated extract. The residue from ethyl acetate extraction was further extracted with *n*-butanol and methanol to obtain *n*-butanol and methanol fractionated extracts, respectively. 

### 3.3. Partial Purification by Vacuum Chromatography 

Simple and rapid chromatography was applied for partial purification of the hexane extract of kaffir lime leaves. A 70/55 mm sinter-glass was filled with silica for column chromatography. During the gradual addition of the silica gel, suction with an appropriate pump and tapping outside the sinter, followed by adding of hexane, were performed to obtain a totally level surface. The hexane extract (10 g) premixed with silica gel was evenly added to the top of the silica bed to obtain a level band of the sample. Solvent mixtures with increasing polarity, composed of hexane, dichloromethane, and methanol were added successively to elute the products. Each fraction was collected and then evaporated with a rotary evaporator to dryness. Each dried fraction was weighed, kept in a closed container, and stored at −20 °C. The percent yields of partial purification fractions of hexane extract have been shown in Appendix A.

### 3.4. Purification of Compounds from Active Fraction 

The subfraction No. 9 (F9) (5 g) was separated by medium pressure liquid chromatography (MPLC) (Smart Flash W-Prep2XY, Yamazen, Osaka, Japan) using a Hi-Flash column (20 mm × 75 mm, Yamazen) with a gradient elution of hexane/ethyl acetate of 94:6, 75:25, 73:27, 70:30, and 65:35. The fractions were then purified by gel permeation chromatography (GPC) (LC-9201, Japan Analytical Industry, Tokyo, Japan) using JAIGEL-1H column (20 × 600 mm) and JAIGEL-2H column (20 × 600 mm) eluting with chloroform to ultimately give two pure fractions. The purity of each collected fraction was determined by TLC and ^1^H NMR spectrum. The structure of two pure compounds was characterized by spectroscopic analyses. Phytol was initially identified by using similarity search software (GCMS-QP2010, Shimadzu, Kyoto, Japan) (data not shown). Lupeol was identified by SciFinder^®^ based on proposed molecular formula which was obtained from high resolution mass spectrometry (HRMS) by Fast atom bombardment (FAB)-MS (JMS-700, JEOL, Tokyo, Japan). Optical rotations were measured on a digital polarimeter (P-2200, JASCO, Tokyo, Japan) at the sodium lamp (*λ* = 589 nm) D-line and are reported as follows: [α]^T^_D_ (*c* g/100 mL, solvent). ^1^H and ^13^C NMR spectra were recorded on spectrometer (500 MHz) (JNM-ECA 500, JEOL, Tokyo, Japan) or on a spectrometer (400MHz) (AVANCE III HD 400, Bruker, Billerica, MA). ^1^H NMR spectra are reported as follows: chemical shift (*δ* ppm), multiplicity (s, singlet; d, doublet; t, triplet; q, quartet; m, multiplet; br, broad), coupling constants (*J*) in Hz, integration, and assignments. ^13^C NMR spectra are reported in terms of chemical shift (δ, ppm).

### 3.5. Characterizations of the Pure Compounds

Phytol: colorless wax; *R*_f_ = 0.31 (hexane/ethyl acetate = 5:1); [α]^23^_D_ +1.7 (*c* 0.29, CHCl_3_) [lit. [α]^23^_D_ +5.0 (*c* 0.01, CHCl_3_)] (Park et al., 2004). ^1^H NMR (500 MHz, CDCl_3_) δ 5.41 (1H, t, *J* = 7.0 Hz, H2), 4.15 (2H, d, *J* = 6.8 Hz, H1), 1.99 (2H, t, *J* = 7.2 Hz, H4), 1.67 (3H, s, 20), 1.52-1.00 (19H), 0.87-0.84 (12H, d, *J* = 6.4 Hz H16/17/18/19); ^13^C NMR (125 MHz, CDCl_3_) 140.5, 123.2, 59.6, 40.0, 39.5, 37.6, 37.5, 37.5, 36.8, 33.0, 32.9, 28.1, 25.3, 25.0, 24.6, 22.9, 22.8, 19.8, 16.3; FAB-HRMS (*m*/*z*) calcd for C_20_H_37_O [M − H]^+^ 295.3001, found 295.2982.

Lupeol: a white solid; *R*_f_ = 0.32 (hexane/ethyl acetate = 5:1); [α]^25^_D_ +31.8 (*c* 0.19, CHCl_3_) [lit. [α]^22^_D_ +27.63 (*c* 3.93, CHCl_3_)] (Swift and Walter, 1942). ^1^H NMR (500 MHz, CDCl_3_) δ 4.68 (1H, brs, H29b), 4.57 (1H, brs, H29a), 3.18 (1H, dd, *J* = 4.9, 11.3 Hz, H19), 2.34-2.41 (1H, m), 1.03 (3H, s, H23), 0.97 (3H, s, H28), 0.94 (3H, s, H26), 0.83 (3H, s, H24), 0.79 (3H, s, H27), 0.76 (3H, s, H25); ^13^C NMR (125 MHz, CDCl_3_) δ 151.1, 109.5, 79.2, 55.5, 50.6, 48.5, 48.1, 43.2, 43.0, 41.0, 40.2, 39.0, 38.9, 38.2, 37.3, 35.7, 34.4, 30.0, 28.1, 27.60, 27.57, 25.3, 21.1, 19.5, 18.5, 18.2, 16.3, 16.1, 15.5, 14.7; FAB-HRMS (*m*/*z*) calcd for C_30_H_50_O [M]^+^ 426.3862, found 426.3861. 

### 3.6. Leukemic Cells and Culture Conditions 

K562 (chronic myelocytic leukemia), Molt4 (lymphocytic leukemia), U937 (monocytic leukemia), and HL60 (promyelocytic leukemia) were used in this study. They were cultured in an RPMI-1640 medium (GIBCO-BRL) containing 1 mM l-glutamine, 100 U/mL penicillin, and 100 µg/mL streptomycin, and supplemented with 10% fetal bovine serum (FBS) at 37 °C under an atmosphere of 95% air and 5% CO_2_.

### 3.7. MTT Cytotoxicity Assay

The MTT (3-(4,5-dimethylthiazol-2-yl)-2,5-diphenyltetrazolium bromide) assay was performed for detecting the cytotoxicity of kaffir lime leaf extracts and purified compounds on leukemic cells. Each strain of the leukemic cells was adjusted to 1.0 × 10^4^ cells in 100 µL of complete RPMI-1640 medium and then added into each well of a flat-bottomed 96-well plate. The cells were incubated at 37 °C under 5% CO_2_ atmosphere, overnight. Various concentrations (3.125 µg/mL to 100 µg/mL) of kaffir lime leaf extracts and (0.488 ng/mL to 1000 ng/mL) of chemotherapeutic drugs (doxorubicin and idarubicin) dissolved in 100 µL of the medium and the medium with and without DMSO, used as vehicle and cell control, were added into each well and incubated for 2 days. Then, 100 µL of the medium were removed and 15 µL of the MTT dye solution (Sigma-Aldrich, USA) were added, and the cells were further incubated for 4 h. After the supernatant was removed, 200 µL of DMSO were added to each well, and then mixed thoroughly to dissolve the formazan crystals. The optical density was measured using an ELISA plate reader at 540 nm with reference wavelength at 630 nm. The percentage of cell survival was calculated from the values of absorbance of the test well and the control wells, using the following Equation (1).
(1)$$\%\;\mathrm{Cell}\;\mathrm{viability}=\frac{\mathrm{Mean}\;\mathrm{absorbance}\;\mathrm{in}\;\mathrm{test}\;\mathrm{well}}{\mathrm{Mean}\;\mathrm{absorbance}\;\mathrm{in}\;\mathrm{vehicle}\;\mathrm{control}\;\mathrm{well}}\times 100$$

The average values of the percentages of cell survival at each concentration obtained from triplicate experiments were plotted as a dose-response curve. The inhibitory concentration at 50% growth (IC_50_) of each extract was determined as the lowest concentration which could inhibit cell growth by 50% compared to the untreated culture, and IC_20_ value of each e extract was determined as the non-toxic concentration to be used for the study of the *WT1* gene expression and the WT1 protein expression.

### 3.8. Total RNA Extraction

After being treated with the extracts, the leukemic cells were harvested and washed three times with ice-cold sterile PBS, pH 7.4, after which the cells were counted for cell viability using 0.2% trypan blue dye. Afterward, the cell pellet was collected and resuspended in 200 µL of sterile PBS, pH 7.4. Then, the total RNA was extracted using High Pure RNA Isolation Kit (Roche, Germany). RNaseOUT™ Recombinant Ribonuclease Inhibitor (Invitrogen™, USA) was added into the microcentrifuge tube with the eluted total RNA for RNA protection. The total RNA was used directly in real-time RT-PCR. The total RNA was determined for purity by measuring the optical density via spectrophotometry at a ratio of 260 nm/280 nm.

### 3.9. cDNA Synthesis

After the determination of the total RNA concentration, cDNA was synthesized using a Transcriptor High Fidelity cDNA Synthesis Kit (Roche, Germany). A template-primer mixture was prepared by adding 0.5 µg of total RNA. The template-primer mixture was denatured by heating the tube for 10 min at 65 °C to ensure the denaturation of RNA secondary structures. The cDNA synthesis was started at 55 °C for 30 min, and the inactivation of reverse transcriptase was performed by heating to 85 °C for 5 min.

### 3.10. Real-time RT-PCR

Real-time PCR analysis was performed using the DyNAmo™ qPCR Kit (Finnzymes, Finland), reagent for quantitative real-time analysis of DNA samples using probe base detection, and Tagman^®^ hydrolysis probe (Operon, USA), based on the hot start Thermos brockianus (Tbr) DNA polymerase. As for the WT1 primers, the forward primer (5′GATAACCACACAACGCCCATC3′), the reverse primer (5′CACACGTCGCACATCCTGAAT3′), and the WT1 probe (5′FAM-ACACCGTGCGTGTG- TATTCTGTATTGG-TAMRA3′) were used. As for the β-actin primer, the forward primer (5′CCCAGCACAATGAAGATCAAGATCAT3′), the reverse primer (5′ATCTGCTGGAAGGTGGA- CAGCGA’3), and the β-actin probe (5′FAM-TGAGCGCAAGTACTCCGTGTGGATCGGCGTA- MRA3′) were used. Due to the large difference in the copy numbers of β*-actin* and *WT1* in the sample, in order to prevent β*-actin*, which had a larger copy number, from using all the PCR components (DNA polymerase, MgCl_2_, dNTP, and dUTP) in the PCR amplification process, the detection of the *WT1* gene expression and the β*-actin* gene expression had to be done in separate tubes. The real-time cycling condition was started at an initial denaturation temperature of 95 °C for 10 min. This step was needed to activate the hot start Tbr DNA polymerase and to denature the template cDNA. After that, PCR amplification was performed for 50 cycles of denaturation at 95 °C for 30 s, an annealing extension at 63 °C for 60 s. The standard curves of WT1 and β-actin were then obtained by making serial dilutions of the K562 cell line cDNA for the validation of the 2^−ΔΔCT^ method. A plot of the log cDNA dilution versus ΔCT was made. If the absolute value of the slope is close to zero, then the efficiencies of the target and reference genes are similar, and the ΔΔCT calculation for the relative quantification of the target may be used.

### 3.11. Protein Extraction and Western Blotting

The whole protein extracts from the treated cells were prepared using RIPA buffer (50 mM Tris-HCl, 150 mM NaCl, 1% Triton X-100, 0.5 mM EDTA, 0.1% SDS, and a protease inhibitor cocktail). The protein concentration was measured using the Folin-Lowry method. Protein was separated by 12% SDS-PAGE and then transferred to PVDF membranes. Membranes were blocked in 5% skim milk and probed by rabbit polyclonal anti-WT1 IgG (Santa Cruz Biotechnology, CA, USA), anti-polyclonal anti-p53 IgG (Santa Cruz Biotechnology, CA, USA), mouse polyclonal anti-cyclin B IgG (BD Transduction Laboratories™, USA), and rabbit polyclonal anti-human GAPDH IgG (Santa Cruz, CA, USA) at dilution 1:1000, 1:1000, 1:500, and 1:1000, respectively, The reaction was followed by HRP-conjugated goat anti-rabbit IgG (Invitrogen™, CA, USA) at 1:10,000 dilution and HRP-conjugated anti-mouse IgG (Promega, WI, USA) at 1:10,000 dilution, followed by HRP-conjugated goat anti-rabbit IgG with the dilution of 1:20,000. The antibody-bound proteins were detected by using the SuperSignal^®^ West Pico Chemiluminescent Substrate (PIERCE, USA), which included two substrate components. The membranes were immediately placed onto a film cassette and then exposed on a clear blue X-ray film (Thermo Fisher Scientific, USA). Finally, the protein band signal was quantified by using a scan densitometer (BIO-RAD, USA).

### 3.12. Cell Cycle Analysis by Flow Cytometry

The subfraction No. 9 (F9) was tested for its effect on cell cycle progression by flow cytometry. After F9 treatments at 3 concentrations at IC_20_ values for 12, 24, 36, and 48 h at 37 °C with 5% of CO_2_, cells were collected, then centrifuged at 1500 × g for 5 min. Cells were washed 3 times in 10 mL ice-cold PBS. During the last centrifugation, the cell pellet was resuspended in 300 µL of PBS, pH 7.4, and then the cells were fixed with 700 µL ice-cold absolute ethanol for 30 min. The cells were then centrifuged at 1500 × *g* for 5 min, and the supernatant was discarded. Then, PI solution (300–500 µL) was added and the cell cycle distribution was examined by flow cytometry (Cytomics FC 500, Beckman Coulter). The result was analyzed using the software program FlowJo V10.

### 3.13. Intracellular Signaling Array

The intracellular signaling pathway involved in leukemic cell growth inhibition after F9 treatment was determined using a PathScan^®^ Intracellular Signaling Array Kit (Chemiluminescent Readout).

### 3.14. Statistical Analysis

All data are expressed as the mean ± standard deviation (SD) or the mean ± standard error of mean (SEM) from triplicate samples of three independent experiments. The statistical differences between the means were determined using one-way ANOVA. The differences were considered significant when the probability value obtained was found to be less than 0.05 (*p* < 0.05).

## 4. Conclusions

The present study demonstrates that the extract of kaffir lime leaves has inhibitory effects on leukemic cell lines. Among the extracts from various solvents, the ethyl acetate and hexane extracts possess the greatest inhibitory effects, and after purification, two active compounds can be isolated and identified as phytol and lupeol. The bioassay of phytol and lupeol against leukemic cells confirmed its antiproliferative activities. These results suggest that phytol and lupeol can be used as chemotherapeutic drug models and potential chemo-therapeutic agents for human leukemia treatment in the future. Although cytotoxicity of phytol and lupeol are less than chemotherapeutic drugs, we believe that these natural products containing in dietary plants will be safer for leukemia patients due to the lower side effects.

## Figures and Tables

**Figure 1 molecules-25-01300-f001:**
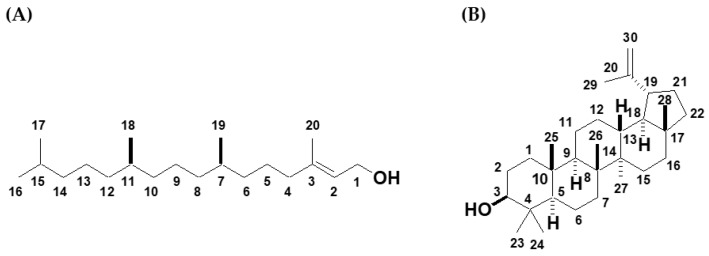
Chemical structures of (**A**) phytol and (**B**) lupeol.

**Figure 2 molecules-25-01300-f002:**
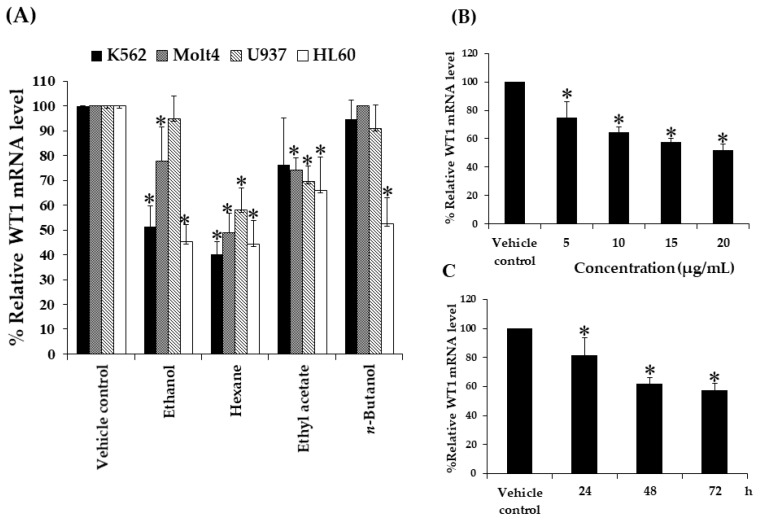
Effects of kaffir lime leaf extracts at IC_20_ on Wilms’ tumor 1 (WT1) mRNA levels in K562, Molt4, U937, and HL60 cells at a density of 1.0 × 10^5^ cells/mL. The cells treated for 2 days in (**A**) complete Roswell Park Memorial Institute (RPMI)-1640 medium, (**B**) the medium with 5, 10, 15, and 20 µg/mL of hexane extract, 0.08% DMSO was used as vehicle control. (**C**) The cells treated for 24, 48, and 72 h, 0.05% with 13.6 µg/mL of hexane extract, DMSO treated for 3 days was used as vehicle control. WT1 mRNA was determined using real-time PCR and presented as %WT1 mRNA levels. The expression levels of WT1 mRNA were normalized using the expression of β-actin mRNA as the internal control. Data are expressed as mean ± SD of three independent experiments. Asterisk (*) demonstrates statistically significant difference between the values of the vehicle control and the sample (*p* < 0.05).

**Figure 3 molecules-25-01300-f003:**
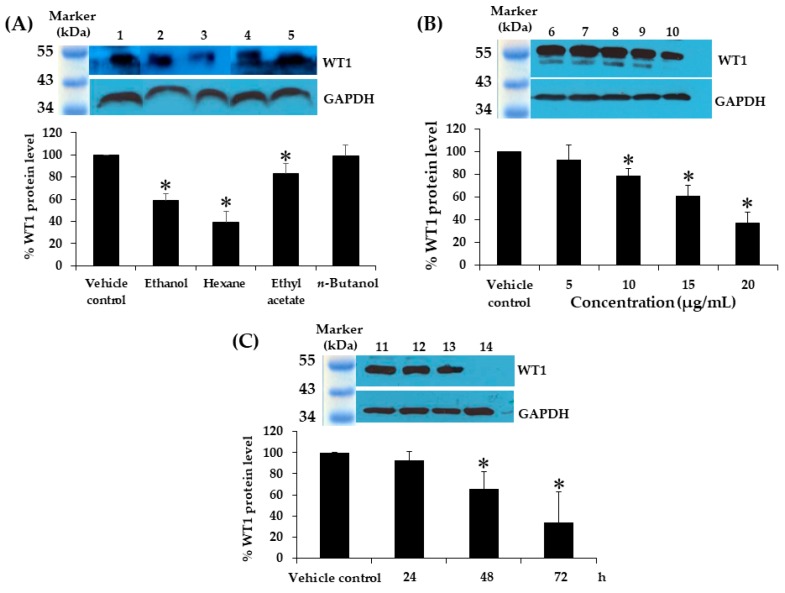
Effects of kaffir lime leaf extracts on WT1 protein levels in K562 cells. (**A**) WT1 and glyceraldehyde 3-phosphate dehydrogenase (GAPDH) proteins following the treatment of (No. 1) 0.18% DMSO (vehicle control) and (No. 2) ethanol, (No. 3) hexane, (No. 4) ethyl acetate, and (No. 5) *n*-butanol extracts at IC_20_ for 48 h. (**B**) K562 cells grown in a medium with (No.6) 0.08% DMSO (vehicle control), and (No. 7) 5 µg/mL, (No. 8) 10 µg/mL, (No. 9) 15 µg/mL, and (No. 10) 20 µg/mL of hexane extract for 2 days. (**C**) K562 cells treated with (No.11) 0.05% DMSO (vehicle control) for 3 days and 13.6 µg/mL of hexane extract for 24 h (No. 12), 48 h (No. 13), and 72 h (No. 14). WT1 and GAPDH protein levels were determined using the Western blot analysis. WT1 (48–54 kDa) and GAPDH (37 kDa) bands were quantified using a densitometer. WT1 protein levels were measured and normalized using GAPDH protein level. Data are expressed as mean ± SD of three independent experiments. Asterisk (*) demonstrates statistically significant difference between the values of the vehicle control and the sample (*p* < 0.05).

**Figure 4 molecules-25-01300-f004:**
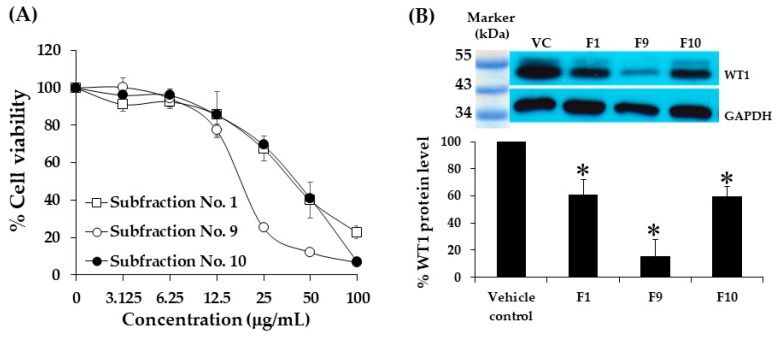
Cytotoxicity and WT1 inhibition of subfraction treatments on K562 cells. (**A**) The cells (1.0 × 10^5^ cells/mL) treated with subfraction numbers 1, 9, and 10, for 48 h. Cell viability was determined using MTT assay. Each point represents mean ± SD of three independent experiments performed in triplicate. (**B**) The cells at a density of 1.0 × 10^5^ cells/mL grown in a medium with 17.8, 10.0, and 16.6 µg/mL of subfractions 1, 9, and 10, respectively, for 48 h, 0.08% DMSO was used as a vehicle control. WT1 and GAPDH protein levels were determined using Western blot analysis, then the WT1 protein levels were measured and normalized using the GAPDH protein level. Data are expressed as mean ± SD of three independent experiments. Asterisk (*) demonstrates statistically significant difference between the values of the vehicle control and the sample (*p* < 0.05).

**Figure 5 molecules-25-01300-f005:**
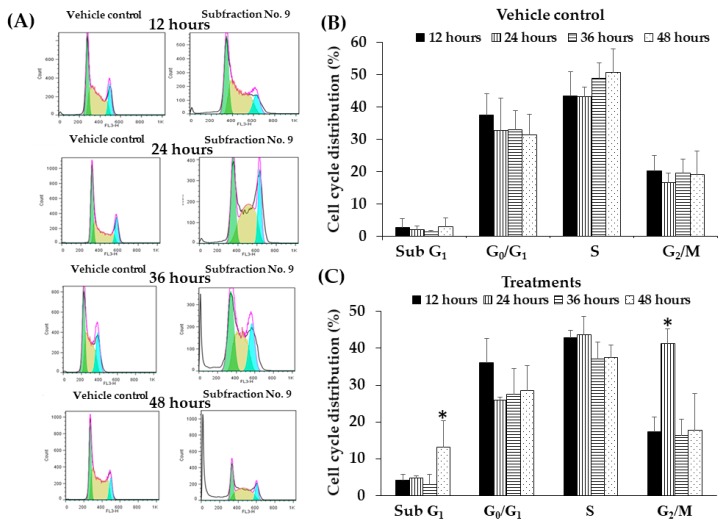
Cell cycle arrested after treated with Subfraction No. 9 (F9) for 12, 24, 36, and 48 h. (**A**) K562 cells at a density of 1.0 × 10^5^ cells/mL treated with 10.0 µg/mL of F9 for 12, 24, 36, and 48 h. Cell cycle was analyzed by flow cytometry after fixation in 75% ethanol and stained with PI. (**B**) and (**C**) Cell cycle distribution after F9 treatments. Data are expressed as mean ± SD of three independent experiments. Asterisk (*) demonstrates statistically significant difference (*p* < 0.05).

**Figure 6 molecules-25-01300-f006:**
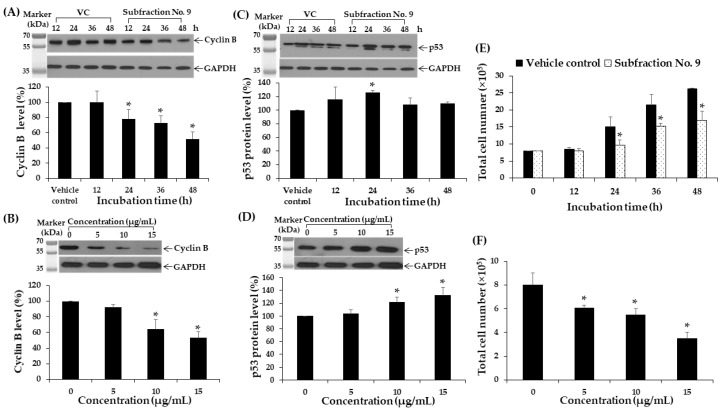
Effects of F9 on cyclin B and p53 proteins in K562 cells. The levels of (**A**) cyclin B after treatment with 10 µg/mL F9 for 12, 24, 36, and 48 h and (**B**) cyclin B after treatment with 5, 10, and 15 10 µg/mL for 48 h. The levels of (**C**) p53 after treatment with 10 µg/mL F9 for 12, 24, 36, and 48 h and (**D**) p53 after treatment with 5, 10, and 15 10 µg/mL for 24 h. Protein levels were assessed by Western blotting. GAPDH was used as a loading control. Protein levels were analyzed with a scan densitometer and normalized using GAPDH protein. (**E**,**F**) Total cell number after treatments with different contact times (12, 24, 36, and 48 h) and concentrations (5, 10, and 15 µg/mL) determined by trypan blue exclusion method. Data are expressed as mean ± SD of three independent experiments. Asterisk (*) denotes a significant difference from the vehicle control group (*p* < 0.05).

**Figure 7 molecules-25-01300-f007:**
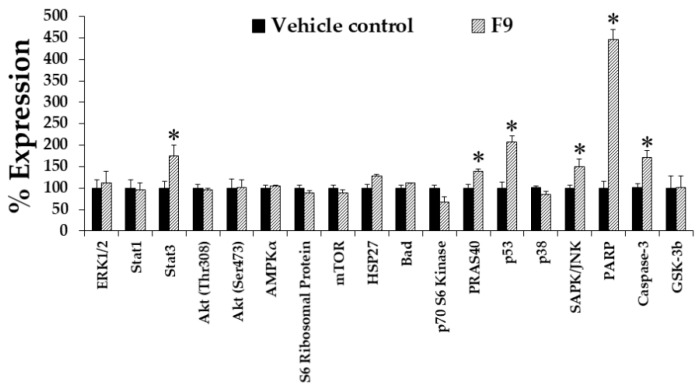
Effects of F9 on phosphor-kinase protein expressions in K562 cells after treating with 23.3 µg/mL for 48 h using a human phospho-kinase array kit. Asterisk (*) demonstrates statistically significant difference between the values of the vehicle control and the sample (*p* < 0.05).

**Figure 8 molecules-25-01300-f008:**
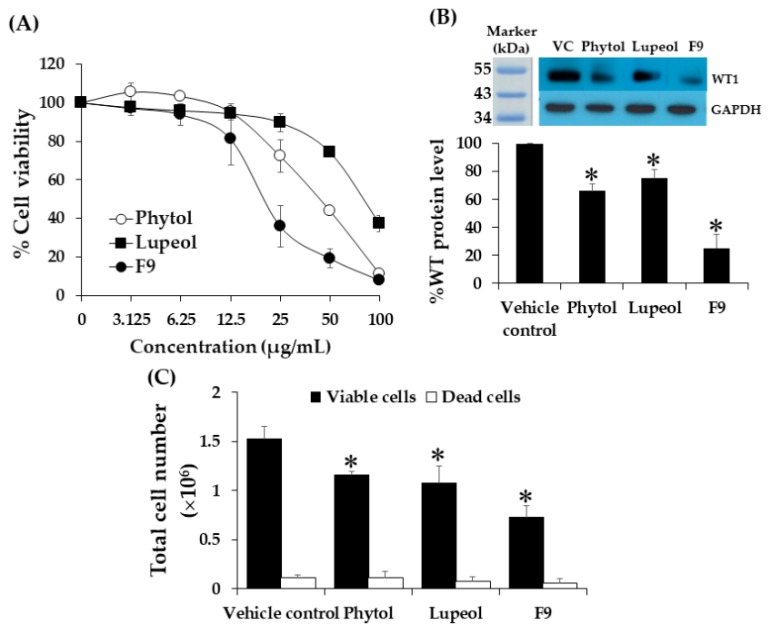
Effects of phytol and lupeol on cytotoxicity, WT1 expression, and total cell number of K562 cells. (**A**) K562 cells at a density of 1.0 × 10^5^ cells/mL treated with various concentrations of phytol and lupeol for 48 h. Cell viability was determined using MTT assay. Each point represents mean ± SD of three independent experiments performed in triplicate. (**B**) The cells at a density of 1.0 × 10^5^ cells/mL grown in a medium with 40 and 21 µg/mL of phytol and lupeol, respectively for 48 h in comparison with and F9 (23.3 µg/mL). The vehicle control was 0.08% DMSO in the medium. WT1 and GAPDH protein levels were determined using Western blot analysis. The WT1 protein levels were then measured and normalized using the GAPDH protein level. (**C**) Total cell number after treatment with phytol and lupeol determined by trypan blue exclusion method. Data are expressed as mean ± SD of three independent experiments. Asterisk (*) demonstrates statistically significant difference between the values of the vehicle control and the sample (*p* < 0.05).

**Table 1 molecules-25-01300-t001:** IC_20_ values of crude kaffir lime leaf fractional extracts determined from plot of percent cytotoxicity on K562, Molt4, U937, and HL60 cell lines.

Crude Kaffir Lime Leaf Fractional Extracts	IC_20_ (µg/mL) (Mean ± SD)			
	K562	Molt4	U937	HL60
Ethanol	40.9 ± 1.3	25.0 ± 3.8	10.2 ± 1.4	17.0 ± 5.8
Hexane	13.6 ± 6.9	2.8 ± 4.7	2.8 ± 3.1	3.7 ± 2.0
Ethyl acetate	11.9 ± 4.4	8.0 ± 0.6	3.2 ± 2.3	8.8 ± 1.4
*n*-Butanol	45.4 ± 1.6	37.5 ± 2.0	18.2 ± 5.3	25.0 ± 4.6

The data is shown as the mean ± SD of three independent experiments.

**Table 2 molecules-25-01300-t002:** IC_50_ and IC_20_ values (µg/mL) of doxorubicin and idarubicin in K562 cell line.

Drug.	IC_50_ Value (Mean ± SD) (µg/mL)	IC_20_ Value (Mean ± SD) (µg/mL)
Doxorubicin	0.80 ± 0.06	0.36 ± 0.04
Idarubicin	0.41 ± 0.04	0.09 ± 0.02

The data are shown as the mean ± SD of three independent experiments.

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
