# Peer review of "Antileukemic Cell Proliferation of Active Compounds from Kaffir Lime (Citrus hystrix) Leaves"

_molecules, 2020, doi:10.3390/molecules25061300_

Round 1

Reviewer 1 Report

The manuscript describes the activity of the extracts and two compounds isolated from  Citrus hystrix leaves. However, several points should be revised.

Abstract: Mentioning popular use right in the abstract indicates that it is important for research and, therefore, the purpose for which the plant is used should be added. Check if the term bioassay-guided fractionation applies to the experimental design of the work. In the phrase “The bioassays confirmed its antiproliferative activities, possibly in a synergistic fashion.” refers to which of the compounds?

Many sentences should be revised or even removed from the text, since they deal with information that is already well disseminated or become repetitive throughout the text, such as:  “The variation in the amount of each component depended on several parameters, such as the region in which the plants grow, the age or vegetative stage of the plant, the storage  condition, and the extraction method [6].”, which describes a widely known and expected behavior for essential oils. The phrase “With regard to cancer research, the bioactive compound in many kinds of citrus fruits is capable of inhibiting cancer cell proliferation.”, it is obvious and unnecessary. Although they are the result of different works, in this sentence below the results are close related and should be described together, to make the discussion less repetitive: “The crude ethanol extracts of kaffir lime leaf and peel showed cytotoxic 87 effects on K562, Molt4, U937, and HL60 cell lines, and the crude kaffir lime leaf extract demonstrated  strong cytotoxic effects on human leukemic cell lines, and were non-toxic to normal PBMCs [15]. Hinoi et al. reported that that kaffir lime leaf and peel ethanol fractionated extracts have cytotoxic effects on K562, Molt4, U937, and HL60 leukemic cell lines [16].”

The sentence “Taken together, the essential oil and extracts from kaffir lime, especially the leaf, are probably a potential source of anticancer agents. However, the difference in methods and solvents used in kaffir lime extraction resulted in the differences of the contents in extracts, including the amounts and types of the active compounds. This also resulted in differences in the biological effects of crude kaffir lime leaf in each fractional extract. ” must be better discussed specifically for cases involving the plant under study, or it becomes obvious and unnecessary.

The name of the all compounds cited in this sentence are misspelled: “The two glyceroglycolipids, l,2-di-o-a-linolenoyl-3-o-galactopyranosyl-sn-glycerol and a mixture of two  compounds, l-o-a-linolenoyl-2-o-palmitoyl-3-o-galactopyranosyl-sn-glycerol and its counterpart, extracted from kaffir lime leaves, were found to be potent inhibitors of tumor promoter-induced  Epstein-Barr virus (EBV) activation and 12-o-tetradecanoylphorbol 13-acetate (TPA), a skin  carcinogen, activities in mice [13].”

“The highest yield was found to be the ethanol extract. It was considered that ethanol can dissolve many compounds present in kaffir lime leaves.” Did the authors expect anything different from the result obtained in item 2.1?

What is the reason why the authors use IC20 and not IC50, to describe the results?

“Possibly, the active  compounds in the ethyl acetate fractionated extract may be different from those found in the hexane  extract,…”. For the authors, would it be possible that the chemical composition of these extracts obtained from successive extraction and with solvents with different extraction capacity could be the same?

the authors should be clear about the bioaccumulation and biotransformation process, because the first part of the title sentence sounds like a compound totally produced by the bacteria: “Tert-butyl phenolic derivatives from Paenibacillus odorifer – a case of bioaccumulation and biotransformation from bacteria.”, and it is not the case. The same suggestion for the abstract: In the abstract, there is no mention of the origin of a precursor from the plastic flasks and it is not clear the process of bioaccumulation and biotransformation to afford compound 1 – these finding are the main result of these research. The scientific names in the title and keyword are not written properly.

Nomenclature of compound 1: It is necessary to standardize the nomenclature of compound 1 and its numbering used in the structural elucidation and in the tables. 

“It has been reported to have cytotoxic effects on solid  tumor cell lines i.e. MCF-7, MDA-MB-231, HeLa, PC3, HT-29, A-549, Hs294T, and MRC-5 with IC50 values ranging from 8.79 to 124.84 μM [31]” The author should make it clear to which compound the phrase refers to.

It is crucial to add the chemical profile of Phase 9 before discussing the activity and possible synergistic effect of the phytol and lupeol.

Detailed information from the chemical analysis and fractionation of the compounds must be added for the experimental procedure to be reproducible. Important information is missing, such as the type of silica particle used.

There are errors in the bibliographic references that must be revised.

Reviewer 2 Report

The authors present a comprehensive research on the active ingredients of kaffir lime including extraction, isolation and structure identification and carried out an exhaustive investigation about the molecular mechanism of its WT1 mediated antileukemic activity. This study provided more in-depth and comprehensive understanding of kaffir lime; however, I have several minor concerns regarding the description and the implementation of the approaches.

# Introduction

There are enormous number of pathways and genes involved in leukemia, WT1 is just one of them. Why did you choose it for research? What significant features makes it stand out from other leukemia-related gene?

# Results and Discussion

The authors used very little space to introduce the extraction, isolation and structural identification of the active ingredients of kaffir lime. I think a detailed introduction will give a better understanding of the physical properties of the ingredients he got, such as the polarity and solubility.

The author has carried out an exhaustive investigation about the antileukemic activity of kaffir lime, but only compared it with vehicle control. A comprehensive comparison between kaffir lime and classic approved drugs for leukemia could provide evidence for its clinical possibility. Related information could be acquired from Drugbank (Nucleic Acids Res. 2018, 46: D1074-D1082) and TTD (Nucleic Acids Res. 2020, 48: D1031-D1041).

# Conclusion and Prospect

Authors should discuss more about their views on the future and prospect of kaffir lime in the field of leukemia treatment.

Reviewer 3 Report

Anuchapreeda et al evaluated the anticancer effect of active compounds isolated from kaffir lime (Citrus hystrix) leaves. Study was interesting and presented with appropriate evidence. Manuscript still need following corrections.

  1. Abstract is not well written. Continuity and flow are missing. E.g.: First two sentence about kaffir lime, again third one about kaffir lime compounds.
  2. Why IC50 was not used for crude extract. Rationale for using IC20 for crude extract.
  3. Include IC50 value in the results. Include supplementary table 1 results in manuscript (not as supplementary)
  4. In figure 6, for Cyclin B time point was 48 h and p53 time point was 24 h (as per figure legend), why?

Minor comments:

  1. Figure legends for Figure 6 was not clear. Rewrite it.
  2. Use uniform time point. Eg: 2 days treatment or 48 h treatment.

Round 2

Reviewer 1 Report

The authors have relevant experimental data on the evaluation of the antileukemic activity of the hexane extract and its fractions obtained from the leaves of the species Citrus hystrix. However, the chemical and phytochemical parts are not properly written. The arguments used to justify the conflicting results obtained in a previous study, whose most active  extract of the same plant was ethyl acetate, are superficial and only undermines the good data from the biological evaluation (lines 156 to 159). This distinct distribution of metabolites according to polarity is expected for hexane and ethyl acetate extracts, after all, that is exactly why different solvents are used. The problem is in arguing that the chemical profile between one study and the other was different without comparing the chemical data and profiles of the two extracts/phases in both works, making the discussion superficial. Another critical part is the conclusion, in which the authors mention the possibility of a synergistic effect of the two molecules isolated from the bioactive fraction 9. However, the chemical profile of fraction 9, obtained by CG-MS, points to more than 15 compounds in its composition, and the isolate compounds are not even the major constituents.

For this reason, the authors should focus their efforts on describing the evaluation of the activity of hexane extract from the leaves of the plant under study, and its bioactive fractions, removing the chemical/phytochemical part. It is worth mentioning the isolation of the two compounds, but suggesting a synergistic effect based on these results is superficial and, again, only harms the other relevant data.

References 1,2,3,5,7,10,11,25,29,30 contain misspellings. Pay attention to the scientific names of the species.

A review of English language is also required.

Reviewer 3 Report

Manuscript can be accepted for the publication.

Author Response

Manuscript can be accepted for the publication.

Response: Thank you very much.

Sincerely yours,

Assoc. Prof. Dr. Songyot Anuchapreeda